# COMBINATORIAL CREATIVITY: A NEW FRONTIER IN GENERALIZATION ABILITIES

## ABSTRACT

Artificial intelligence (AI) systems, and Large Language Models (LLMs) in particular, are increasingly employed for creative tasks like scientific idea generation, constituting a form of generalization from training data unaddressed by existing conceptual frameworks. Despite its similarities to compositional generalization (CG), combinatorial creativity (CC) is an *open-ended* ability. Instead of evaluating for accuracy or correctness against fixed targets, which would contradict the open-ended nature of CC, we propose a theoretical framework and algorithmic task for evaluating outputs by their degrees of *novelty* and *utility*. From here, we make several important empirical contributions: (1) We obtain the first insights into the scaling behavior of creativity for LLMs. (2) We discover that, for fixed compute budgets, there exist optimal model depths and widths for creative ability. (3) We find that the *ideation-execution gap*, whereby LLMs excel at generating novel scientific ideas but struggle to ensure their practical feasibility, may be explained by a more fundamental *novelty-utility tradeoff* characteristic of creativity algorithms in general. Though our findings persist up to the 100M scale, frontier models today are well into the billions of parameters. Therefore, our conceptual framework and empirical findings can best serve as a starting point for understanding and improving the creativity of frontier-size models today, as we begin to bridge the gap between human and machine intelligence.

## 1 INTRODUCTION

Einstein famously remarked that "Combinatory play seems to be the essential feature in productive thought," (Hadamard, 1954) referring to the cognitive processes he believed underpinned creative insight in mathematics and the sciences. Indeed, there is a rich body of literature that models creativity as a combinatorial process in the space of mental representations (Koestler, 1964; Boden, 2004; Simonton, 2004; 2021). In the cognitive sciences, Boden (2004) distinguishes between three forms of creativity, of which *combinatorial creativity*—the generation of novel ideas by making unfamiliar combinations of familiar concepts—has played a well-documented role in scientific discovery, technological innovation, and artistic pursuits throughout history (Thagard, 2012; Simonton, 2010). From the invention of the printing press to Darwin's theory of natural selection, the act of connecting previously unrelated concepts has historically been a cornerstone of progress (Koestler, 1964; Eppe et al., 2018; Fauconnier and Turner, 2008).

We now attempt to employ AI systems in scientifically creative tasks once conceptualized by Einstein (Gu and Krenn, 2024; Si et al., 2024; Sanyal et al., 2025), yet they lack strong mathematical and conceptual foundations for the abilities underlying these tasks. As a result, many problems have surfaced. LLM-generated ideas for scientific discovery often suffer from practical infeasibility, make unrealistic assumptions, and omit proper baselines, leading to what has been termed the *ideation-execution gap* (Si et al., 2025). Without a foundational understanding of creativity, our ability to diagnose and improve the outcomes of LLMs for such tasks remains severely limited.

To address these limitations in a controlled way, we introduce a formal framework and an open-ended, algorithmic task for evaluating combinatorial creativity. Our framework models creativity within a conceptual space represented as a large synthetic graph, where models must find novel paths between concepts while adhering to logical constraints. We use this as a minimal testbed that isolates structural aspects of creative generalization. Within this setting, we conduct a systematic empirical study of

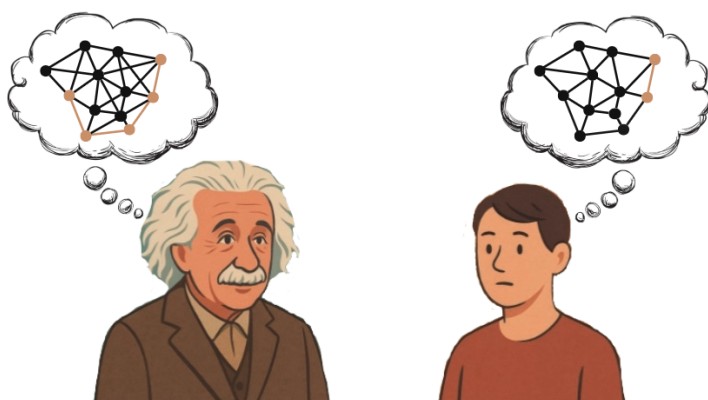

Figure 1: **Combinatorial creativity and cognitive associations.** Since the seminal work of Mednick (1962), creative ability among humans has long been associated with richer associative hierarchies (Simonton, 2004) believed to enable the realization of combinations of distant representations (Thagard, 2012; Simonton, 2021; Koestler, 1964) that leads to breakthrough discovery.

decoder-only Transformers, varying their size, depth, and width across 1M–100M parameters and training compute budgets to probe how these choices relate to creative performance.

First, we obtain initial evidence about the scaling behavior of combinatorial creativity, observing predictable improvements in performance with increased model size and training compute within our parameter regime. Second, we uncover an architectural trend: for a fixed computational budget on this task, wider, shallower models outperform deeper, narrower ones, with an intermediate depth–width tradeoff that maximizes creativity. Third, we perform a detailed error analysis, which reveals that as task complexity increases, models more often fail by violating utility constraints than by producing trivially non-novel outputs. Finally, we empirically recover a fundamental *novelty–utility tradeoff* predicted by prior theory (Varshney, 2019); in our experiments this tradeoff remains pronounced across all model sizes studied. These results do not aim to characterize the creative limits of frontier models but instead provide a controlled, algorithmic instance of phenomena—such as the tension between novelty and feasibility—that have been observed in scientific ideation with LLMs. Together, our conceptual framework and empirical findings offer a starting point for studying and improving the creativity of modern AI models, and for extending this line of work to larger scales and more semantically grounded conceptual spaces.

## 2 BACKGROUND

### 2.1 BACKGROUND ON CREATIVITY

**Defining Creativity** Creativity is defined as *the generation of novel, useful, and surprising artifacts* (Simonton, 2010; 2021; Boden, 2004; Varshney, 2019; Schapiro et al., 2025; Sanyal et al., 2025). Though creativity can refer to a person, process, product, or press (environment) (Rhodes, 1961), in the study of computationally creative systems, it is most common to adopt the product or process view (Varshney, 2019). Moreover, in this case, it is also convenient to consolidate novelty and surprise into one dimension (Varshney, 2019), which we hereafter refer to simply as *novelty*.

**Types of Creativity** Boden (2004) famously distinguishes between three types of creativity: combinatorial creativity (CC), exploratory creativity (EC), and transformational creativity (TC). The first models the creation of new artifacts as combinations of existing elements in a space of possible components. Consider recipe design (Varshney et al., 2019), for example, where new recipes are generated by taking combinations of existing ingredients in varying proportions. The latter types, exploratory and transformational, are historically defined with respect to a "conceptual space," a set of rules and constraints that defines what constitutes well-defined and intelligible artifacts in a particular

domain. Exploratory creativity refers to artifacts generated by following these rules and constraints (such as AlphaGo move 37 (Silver et al., 2017)) whereas transformational creativity, which refers to the more difficult task of re-structuring the very rules of a conceptual space, is considered the pinnacle form of creativity for its historical role in breakthrough innovation (Boden, 2004). Famous examples of transformational creativity include Einstein's relativity theory, the shift from geocentrism to heliocentrism, and the discovery of air pressure (Haven, 2007; Schapiro et al., 2025; Thagard, 2018; Koestler, 1964).

**Combinatorial Creativity**  The study of combinatorial creativity dates back to Hadamard (1954), which provides a survey of introspective accounts from famous mathematicians, scientists, and even musical composers in which creative ideation is described as a combinatorial process. The French mathematician Henri Poincarè describes one scenario in which "ideas rose in crowds; [he] felt them collide until pairs interlocked, so to speak, making a stable combination" (quoted in Hadamard (1954), p.15). Mednick (1962) later demonstrates that human creativity can be understood as a process of associating or combining mental representations, with more distant associations correlated with more creative artifacts. Based on this finding, Mednick developed the remote association test (RAT) for measuring human creativity. Koestler (1964) later described a combinatorially creative framework named *bisociation*, where discoveries occur when two previously unrelated matrices of thought are suddenly recognized as compatible, in a moment of creative insight. This model is used to account for humor, art, scientific breakthroughs, and technological inventions, ranging from Gutenberg's printing press and Kepler's planetary laws to Darwin's natural selection. Boden (2004) was the first to explicitly define the term combinatorial creativity. Subsequent studies have shown that nearly all of the most impactful scientific discoveries and technological inventions in human history (Haven, 2007) can be modeled as combinatorial (Thagard, 2012; Simonton, 2010; 2021; 2004). This suggests that understanding and improving the combinatorial creativity abilities of AI models can have a significant impact on their ability to engage in scientific and technological discovery.

## 2.2 Distinguishing Combinatorial Creativity from Classical Forms of Generalization

Among the five types of generalization studied in NLP research (Hupkes et al., 2022), *combinatorial creativity* (CC) most closely resembles *compositional generalization* (CG). Broadly, compositionality is a linguistic principle that the meaning of a complex expression is a function of the meaning of its parts and the way they are combined (Kim and Linzen, 2020; Fodor and Pylyshyn, 1988). CG is divided into one of five types: (i) systematicity, (ii) productivity, (iii) substitutivity, (iv) localism, and (v) overgeneralization (Sinha et al., 2024; Hupkes et al., 2020). For a full survey on CG, see Sinha et al. (2024) and Lin et al. (2023).

**Aspects of Comparison**  In Table 1, we compare generalization abilities along six key aspects. An ability is *compositional* (A1) if it involves recombination of atomic units into compound artifacts; *open-ended*[*] (A2) if there is no single correct answer for its evaluation, but instead multiple plausible answers; *structurally novel* (A3) if it generates artifacts whose form is distinct from structures trained on; and *semantically novel* (A4) if generated artifacts have new meanings. Lastly, an ability involves measuring *degrees of novelty* (A5) and *degrees of utility* (A6) if artifacts may be more or less novel or useful, respectively, depending on their semantic or structural properties.

**Systematicity (CG-S)**  Systematicity refers to the ability to systematically recombine known parts and rules (Hupkes et al., 2020; Lake and Baroni, 2017; Kim and Linzen, 2020; Li et al., 2019). This is inherently compositional (A1), structurally novel (A3), and semantically novel (A4). For example, if one has learned the words `black` and `dog` separately, can they compose them together in the expression `black dog`? Popular tests for systematicity involve sequence-to-sequence tasks (Lake and Baroni, 2017; Kim and Linzen, 2020; Li et al., 2019) which evaluate against fixed, ground-truth sequence-to-sequence targets. As a result, systematicity evaluation is not open-ended (A2).

**Productivity (CG-P)**  Productivity refers to the ability for models to extend predictions beyond the length they have seen in their training data (Hupkes et al., 2020; Anil et al., 2022). Clearly, this

---

[*]Note that our notion of open-endedness is slightly different from the recent definition in Hughes et al. (2024) because we consider open-endedness from the **p**roduct, not **p**rocess, perspective (Rhodes, 1961)

Table 1: **Comparison of forms of compositional generalization, productivity (CG-P) and systematicity (CG-S), with combinatorial creativity (CC) along six key dimensions.** (A1) *Compositionality*: all three abilities always construct compositional objects; (A2) *Open-Ended*: CC is the only ability which must always be evaluated in an open-ended way, meaning there are always many ways to adequately solve a particular task; (A3) *Structural Novelty*: CG-P always involves generalizing to unseen lengths and structures, whereas this is only true of CG-S and CC sometimes; (A4) *Semantic Novelty*: CG-S and CC always involve combining primitives in a way that leads to semantically novel structures, whereas this is only true of CG-P sometimes; (A5) *Degree of Novelty* and (A6) *Degree of Utility*: CC is the only ability which always quantifies the novelty and utility of its artifacts in degrees, rather than by binary evaluation. On the right, we compare our framework in Section 3 against sibling discovery (SD) and triangle discovery (TD) from Nagarajan et al. (2025). A more detailed comparison of our framework and SD/TD is given in Section 3.5.

| Aspect | Form of Generalization | | | CC Framework & Tasks | | |
|---|---|---|---|---|---|---|
| | **CG-P** | **CG-S** | **CC** | **SD** | **TD** | ***Ours*** |
| *Compositionality* | ✔ | ✔ | ✔ | ✔ | ✔ | ✔ |
| *Open-Endedness* | ✗ | ✗ | ✔ | ✔ | ✔ | ✔ |
| *Structural Novelty* | ✔ | ✗/✔ | ✗/✔ | ✗ | ✗ | ✗/✔ |
| *Semantic Novelty* | ✗/✔ | ✔ | ✔ | ✔ | ✔ | ✔ |
| *Degree of Novelty* | ✗ | ✗ | ✔ | ✗ | ✗ | ✔ |
| *Degree of Utility* | ✗ | ✗ | ✔ | ✗ | ✗ | ✔ |

involves compositionality (A1) and structural novelty (A3). One example of productivity is whether one could solve 1555 ÷ 171 if taught to perform long division for only two-digit integers, e.g., 82 ÷ 16. Productivity is only sometimes semantically novel (A4): adding or multiplying integers with more digits than those trained on (Zhou et al., 2024) involves generalizing a deterministic algorithm without producing new meanings, whereas understanding or generating sentences that are longer than ones encountered during training (Ahuja and Mansouri, 2024) could involve semantic novelty. Like systematicity, productivity can be evaluated in a closed-ended fashion (A2).

**Combinatorial Creativity (CC)**  Combinatorial creativity is a compositional (A1), open-ended (A2) ability that always involves creating or discovering new meanings in new forms, leading to structural (A3) and semantic (A4) novelty. However, unlike both CG-S and CG-P—which do not measure degrees of novelty (A5) and utility (A6) for open-ended artifacts—existing mathematical theories of CC explicitly define continuous novelty and utility functions that measure the *degree of novelty* and *degree of utility* for creative artifacts (Varshney, 2019; Maher, 2010). We will now introduce a theoretical framework for CC that addresses each of the six aspects previously discussed.

## 3 A Theoretical Framework and Open-Ended, Algorithmic Task for Combinatorial Creativity

We provide a mathematical framework for CC that involves generating open-ended, compositional objects in a fixed conceptual space. Importantly, our framework allows us to controllably measure the novelty and utility of creative artifacts, an integral aspect of evaluation for creativity (Simonton, 2010; Maher, 2010; Varshney, 2019) overlooked by prior task frameworks in Nagarajan et al. (2025). Our algorithmic task prompts models to compose a labeled path between two nodes while obeying *logical constraints* (inclusion/exclusion of edge labels). Evaluation is inherently open-ended: any artifact that satisfies the constraints is valid and can be further evaluated by its degree of novelty and utility.

### 3.1 Combinatorial Creativity Setting

Combinatorial creativity occurs in conceptual spaces, where atomic units (or "concepts") are composed to form combinatorial objects (Boden, 2004; Varshney, 2019). It is common to model concep-

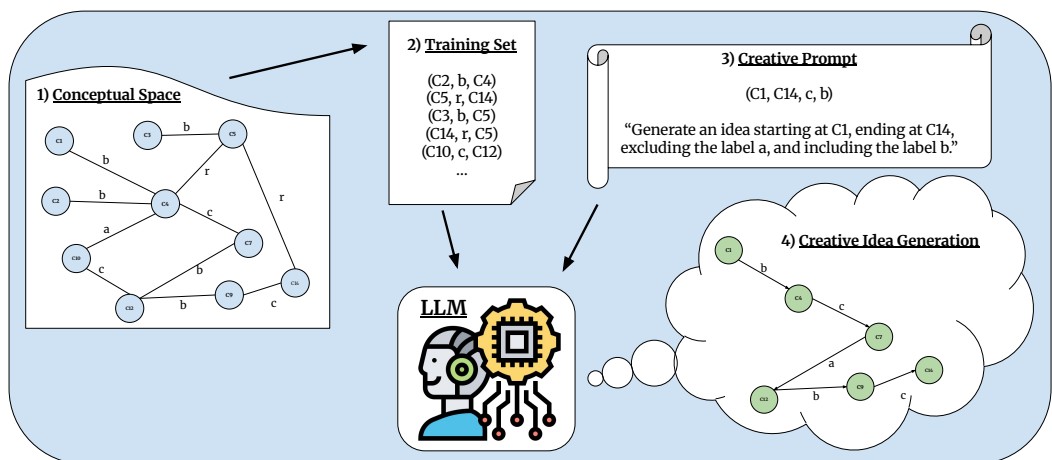

Figure 2: **An open-ended, algorithmic framework for evaluating combinatorial creativity (CC) abilities.** A model is pre-trained on concept-relation-concept triples drawn from an underlying conceptual space. At test-time, creative prompts ask the model to generate "ideas" between distant start and end concepts while adhering to increasing levels of inclusion-exclusion, logical constraints. Idea generation is done fully in-weights, not in-context, since CC involves recalling facts in-memory.

tual spaces as graphs (Thagard, 2018; Schapiro et al., 2025), where nodes represent concepts and edges represent semantic relations between concepts.

**Definition 1** (Conceptual Space). We define a conceptual space as a simple, undirected, and labeled graph $G = (\mathcal{V}, \mathcal{E}, \Sigma)$ with nodes $\mathcal{V}$, labeled edges $\mathcal{E} \subseteq \{\{u, v\} \times \{\ell\}\}$, and lowercase label alphabet $\Sigma = \{a, \ldots, z\}$.

We write $u \overset{\ell}{\leftrightarrow} v$ for the undirected edge $\{u, v, \ell\}$, and define directed adjacency $\mathcal{N}(u, \ell) = \{v : u \overset{\ell}{\leftrightarrow} v\}$. To isolate the study of creativity and prevent the confounding effect of the reversal curse (Berglund et al., 2023), we use undirected edges. We let $\mathbf{w} \in \Delta\Sigma$ denote a non-uniform distribution over edge labels, which will later be used in Definition 4 to calculate novelty. Next, taking inspiration from Varshney et al. (2020), we represent creative artifacts as labeled walks on $G$.

**Definition 2** (Creative Artifact). A creative artifact $P$ is a labeled walk on $G$

$$P = (v_0, \ell_1, v_1, \ell_2, \ldots, \ell_h, v_h), \quad v_t \in \mathcal{V}, \ \ell_t \in \Sigma, \ \text{with } v_t \in \mathcal{N}(v_{t-1}, \ell_t) \ \forall t \in \{1, \ldots, h\}. \quad (1)$$

We let $\mathcal{P}$ denote the space of all possible creative artifacts admissible by Definition 2. From here, creative prompts task models with discovering valid connections between a given pair of concepts, while adhering to inclusion-exclusion constraints that govern the validity of the association. This serves a minimal abstraction of the creative process among humans, which involves making semantically distant associations (Mednick, 1962; Gray et al., 2019).

**Definition 3** (Creative Prompt). A creative prompt is a tuple $x = (u, v, \mathcal{I}, \mathcal{X})$ consisting of (i) a starting concept $u \in \mathcal{V}$, (ii) an ending concept $v \in \mathcal{V}$, (iii) an inclusion set $\mathcal{I} \subseteq \Sigma$ of edges that must be present in the path, and (iv) an exclusion set $\mathcal{X} \subseteq \Sigma$ of edges that must be excluded from the path, such that $\mathcal{I} \cap \mathcal{X} = \emptyset$.

We let $\mathcal{T}$ denote the space of all possible prompts defined according to Definition 3.

### 3.2 QUANTIFYING DEGREES OF NOVELTY

One condition for an artifact to be judged creative is that it must be novel. Given an artifact $P$, there are two common ways to measure its novelty: (i) as some function of the distance $d$ between $P$ and a set of existing artifacts $d(f(P))$ (Maher, 2010), or, for combinatorial creativity especially,

(ii) semantic graph distances induced by the combinatorial components[†] (Varshney et al., 2020; Gray et al., 2019). To keep the algorithmic task as controllable as possible, we adopt method (ii), quantifying *novelty* via the graph walk distance and the surprise of the labels used on the walk, which can be understood as a proxy for semantic distance (Gray et al., 2019).

**Definition 4** (Novelty). Given a non-uniform distribution over edge labels $\mathbf{w} \in \Delta\Sigma$ and a creative artifact $P$ of length $h$, defined according to Definition 2, its novelty is given by:

$$\text{N}(P) := \alpha_h h + \alpha_r S(P) \tag{2}$$

where $S(P) = \frac{1}{k}\sum_{i=1}^{k} -\log(w_{l_i})$ is the surprise of the path, defined as the average negative log-likelihood of the label probabilities $w_{l_i}$ given in Definition 1, and $\alpha_h, \alpha_r > 0$ are controllable, scalar parameters.

## 3.3 QUANTIFYING DEGREES OF UTILITY

In addition to being novel, creative products must also be *useful* in order to be judged creative (Varshney, 2019; Maher, 2010; Boden, 2004; Simonton, 2010). A common way to evaluate utility is to ensure that artifacts obey logical constraints, representing domain-specific rules over what is useful or not (Boden, 2004; Mayer, 1994; Schank and Cleary, 1995; Schapiro et al., 2025). A natural way to operationalize utility, therefore, is as *inclusion and exclusion constraints* over graph walks.

**Definition 5** (Utility). Given a creative artifact $P$ defined according to Definition 2, a set of inclusion constraints $I$, and a set of exclusion constraints $X$ (where $X$ and $I$ are disjoint, i.e. $I \cap X = \emptyset$), the utility of $P$ is given by:

$$\text{U}(P; x) := (1 + \alpha_I|I|)\,(1 + \alpha_X|X|)\,\mathbb{I}[v_0 = u, v_h = v, \{\ell_1, ..., \ell_h\} \supseteq I, \{\ell_1, ..., \ell_h\} \cap X = \emptyset] \tag{3}$$

where $\alpha_I, \alpha_X > 0$ are controllable, scalar parameters.

The utility function consists of three main parts: the terms $(1 + \alpha_I|\mathcal{I}|)$ and $(1 + \alpha_X|\mathcal{X}|)$ scale the utility function in proportion to the number of inclusion and exclusion constraints, respectively, while the indicator term ensures that artifacts obey these constraints and start and end at the correct nodes.

**Evaluation Set Generation**   To create a structured and challenging evaluation set, we generate problems in a level-based hierarchy. This process ensures a controlled distribution of difficulty, primarily organized by path length (hops) and the number of constraints.

First, for each hop count $h \in \{1, \ldots, 6\}$, we generate a fixed number of "base paths" by randomly sampling start and end nodes $(u, v)$ and finding a shortest path between them of exactly length $h$ using a breadth-first search (BFS).

For each base path found, we generate a hierarchy of $L_{\max} = 5$ evaluation instances, or "levels."

- **Level 1:** The query consists of the base path's $(u, v)$ pair with no constraints ($I = \emptyset, X = \emptyset$).
- **Level $l > 1$:** We introduce $l - 1$ constraints. For each constraint, we decide with probability $p_{inc} = 0.5$ to add an inclusion constraint; otherwise, we add an exclusion constraint. Inclusion labels are drawn randomly from the set of labels present in the original base path, while exclusion labels are drawn from the set of labels not present in it. For each of these new constrained queries, a new ground-truth path is found using a constrained BFS that maintains the original hop count $h$. This guarantees that a valid, non-trivial solution exists for every evaluation problem.

This procedure results in a multi-faceted evaluation set where difficulty increases both with path length and the number of active constraints.

## 3.4 MEASURING CREATIVITY

Now, we can provide a continuous measure for evaluating the creativity of an artifact $P$ with respect to a distribution over prompts in a fixed conceptual space. Following Maher (2010) and Simonton (2010), our creativity score is multiplicative in novelty and utility.

---

[†]Note that under certain conditions, semantic graph distances are asymptotically equivalent to statistical distances to existing artifact sets (Varshney et al., 2020)

**Definition 6** (Creativity). Let $G_\theta : \mathcal{T} \to \mathcal{P}$ be a generative model and $\mathcal{D}$ the evaluation distribution over the space of prompts $\mathcal{T}$. The creativity of $G_\theta$ is given by

$$C(\theta) := \mathbb{E}_{x \sim \mathcal{D}} [U(G_\theta(x); x) \cdot N(G_\theta(x))] . \tag{4}$$

### 3.5 DETAILED COMPARISON WITH SIBLING AND TRIANGLE DISCOVERY

We compare our framework with the sibling discovery (SD) and triangle discovery (TD) tasks for combinatorial creativity presented in Nagarajan et al. (2025) along three key aspects from Table 1.

1. **Structurally novel artifacts:** In both SD and TD, test-time artifacts are restricted to the exact form witnessed during training–(sibling, sibling, parent) triples in the case of SD and (edge, edge, edge) triples in the case of TD–and evaluation only probes whether test-time artifacts are semantically novel. While this design choice makes the evaluation more practically convenient, it restricts any form of structural novelty through generalization to unseen lengths, which is a critical aspect of CC. We note that the authors directly concede this limitation, stating they "are looking at a simple form of novelty that is in-distribution" (p. 4). Our creative artifacts do not provide any restriction on length (see Definition 2).

2. **Degrees of novelty:** The algorithmic creativity evaluation in Nagarajan et al. (2025) treats novelty as a binary function (e.g., "was this (sibling, sibling, parent) triple in the training set or not?"), whereas real-world evaluation of creative artifacts requires measuring novelty in degrees (Varshney, 2019; Simonton, 2010; Maher, 2010). In Definition 4, we provide a continuous measure of novelty.

3. **Degrees of utility:** The evaluation of the utility of outputs in Nagarajan et al. (2025) only considers whether outputs are *coherent* (whether or not all the nodes are valid), which fails to fully capture the scope of logical constraints reflective of real-world creative artifacts. We provide a minimal abstraction of real-world, utility criteria by designing two categories of logical constraints: (i) *inclusion constraints*, which require that paths include certain labels, and (ii) *exclusion constraints*, which forbid paths from including certain labels. In Section 5, we explain how these constraints serve as a minimal abstraction of key empirical failure modes observed when LLMs perform scientifically creative idea generation (Si et al., 2024; 2025).

## 4 EXPERIMENTS

**Key Research Questions** We are interested in how fundamental architectural choices influence the creativity of LLMs on the task defined in Section 3. For example, Nagarajan et al. (2025) recently found creative gains from changing the pre-training objective from next-token to multi-token prediction. In this study, we are especially curious how model creativity is impacted by scale and architecture choice

### 4.1 MODEL ARCHITECTURE

We perform experiments on autoregressive language models, based on the GPT-2 decoder-only Transformer architecture (Radford et al., 2019). To obtain a dense "creativity landscape" across architectural space, we perform a multi-dimensional sweep of models at varying parameter buckets of approximately 1 million, 10 million, and 100 million parameters. Within each bucket, we systematically vary the model's depth, width, and number of attention heads to disentangle their impact on creativity. For a detailed explanation of the dataset construction and task implementation, see Appendix B.

**Depth** ($L$) **vs. Width** ($E$)**:** For each parameter bucket, we define a set of aspect ratios. We trade off the number of layers ($L$) against the embedding dimension ($E$) while keeping their product, $L \times E$, roughly constant. This allows us to study whether combinatorial ability is better supported by wider, shallower models (which may excel at representing a vast number of concepts simultaneously) or by narrower, deeper models (which may be better suited for complex, sequential reasoning). The MLP inner dimension is held at a constant multiple of the embedding size ($4 \times E$), following standard practice.

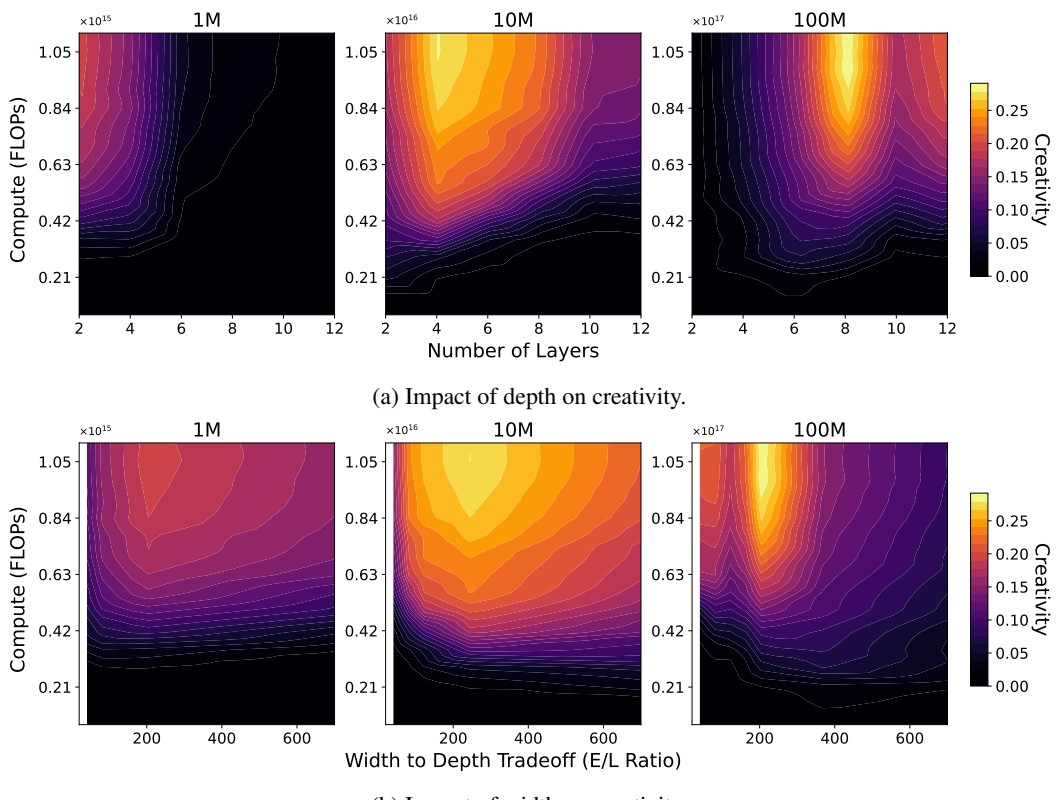

(a) Impact of depth on creativity.

(b) Impact of width on creativity.

Figure 3: **The impact of width and depth on creativity.** These heatmaps visualize the combinatorial creativity of models across three distinct parameter budgets (1M, 10M, and 100M). For each budget, the vertical axis represents the amount of training compute in FLOPs. The color intensity corresponds to the model's creativity score, while the horizontal axis represents the number of layers $L$ (Figure 3a) or the width to depth ratio $E/L$ (Figure 3b). The contours reveal a clear, non-monotonic trend: in Figure 3a, creativity improves as layers are added up to a certain point, after which performance declines, and in Figure 3b, creativity improves as the width is increased up to a certain point, after which performance also declines. The optimal depth becomes more pronounced at larger scales, with the 100M models achieving peak creativity around 8 layers, while the optimal performance for width is at an $E/L$ ratio between 200 and 300.

**Number of Attention Heads ($H$):** For each $(L, E)$ configuration, we further sweep the number of attention heads $H \in \{1, 2, 4, 8, 16, 32\}$, subject to the constraint that $E$ must be divisible by $H$. The number of heads dictates the multiplicity of representational subspaces the model can simultaneously attend to. We hypothesize that a larger number of heads may be critical for managing the multiple, independent constraints present in our combinatorial tasks.

## 5  RESULTS AND DISCUSSION

**The existence of optimal depths and widths for creativity.**  In Figure 3a, we visualize the impact of the number of layers $L$ on combinatorial creativity across all three model sizes. Our most significant finding is that for a fixed parameter count, there is an architectural "sweet spot," an optimal number of layers that maximizes creativity, after which increasing depth further can be detrimental. For the 100M models, this peak is clearly visible around 8 layers. Models that are too shallow (e.g., 2-4 layers) or too deep (e.g., 12+ layers) for their parameter count are substantially less creative. Similarly, in Figure 3b, we visualize the impact of the width-to-depth ratio on the creativity of models at all three scales. Note that when depth is increased within a fixed parameter budget, the model's width (embedding dimension) must necessarily decrease. For a fixed parameter count, there is also

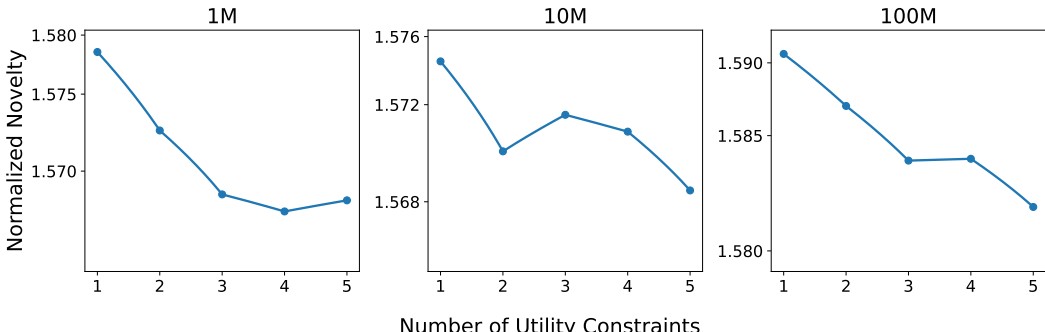

Figure 4: **The novelty-utility tradeoff persists across scales**: These plots show the relationship between the number of utility constraints (x-axis) and the normalized novelty of generated creative artifacts (y-axis) for models of three different parameter scales: 1M, 10M, and 100M. Novelty is normalized by the mean novelty of simple, single-hop paths at each constraint level to isolate the effect of complexity. A clear downward trend is visible across all scales, indicating that as more utility constraints are imposed, the novelty of the generated artifacts tends to decrease.

an optimal width-to-depth ratio that maximizes creativity, after which increasing the width further can be detrimental. The optimal $E/L$ ratio occurs between 200 and 300 for all three model sizes. This suggests that combinatorial creativity requires a delicate balance between (1) models that are *too shallow and wide*, where insufficient depth may hinder the sequential processing capacity to handle in-memory leaps of thought (which are required to make distant, constrained associations between concepts) and (2) models that are *too deep and narrow*, which suffer from restricted representational capacity that may limit their ability to hold and associate the diverse concepts needed for novel combinations. Future work can use our framework as a starting point to explore this depth-width tradeoff in more detail mechanistically.

**The novelty-utility tradeoff.** In Figure 4, we plot the relationship between novelty and utility across all three model sizes. Previously, Varshney (2019) established a fundamental, information-theoretic limit between novelty and utility for combinatorial creativity. We find a similar novelty-utility tradeoff holds here: across all three scales, as the number of utility constraints increases, the novelty of artifacts exhibits a clear downward trend. While this tradeoff does not improve by increasing model size to 100M, frontier models today are well into the billions of parameters. Our work provides a foundation for future studies to explore this tradeoff for billion-parameter models.

**Understanding the ideation-execution gap for LLM-generated ideas.** A series of recent studies have attempted to apply combinatorial creativity explicitly for scientific idea generation (Radensky et al., 2024; Sternlicht and Hope, 2025; Zhao et al., 2025). With the novelty-utility tradeoff in mind, we provide a potential explanation for why LLMs excel at generating novel research ideas (Si et al., 2024; Sanyal et al., 2025; Gu and Krenn, 2024; Wang et al., 2024; Guo et al., 2025) but struggle at ensuring their practical feasibility, in what has been termed the *ideation-execution gap* (Si et al., 2025). In Table 2, we explain how exclusion constraints can be viewed as a minimal abstraction for preventing unrealistic assumptions and excluding prohibitively expensive execution plans, while inclusion constraints can represent ensuring that a proper baseline is included and can serve as a minimal abstraction to ensure implementation plans are sufficiently detailed. Since the novelty-utility tradeoff remains persistent even at the 100M scale (see Figure 4), this suggests that the same fundamental tradeoff might plague the frontier models used in previous works, although a large-scale study pretraining at frontier-model scale should be performed to validate this explicitly. This finding is consistent with recent work from Shashidhar et al. (2025), which also identified a validity-diversity tradeoff in LLM-generated evaluation questions, where models that produced the most diverse (novel) questions often did so at the cost of lower factual validity (utility).

**Isolation of errors** In Figure 5, we plot the distribution of error types among creative artifacts that failed to satisfy the utility predicate in Definition 5. The most common error type is hallucination, in which a model outputs an invalid edge or node. At smaller scales (1M, 10M), hallucinations dominate

Table 2: **Key failure modes of LLMs for scientific idea generation** (Si et al., 2024; 2025; Guo et al., 2025) and mapping of failure mode to inclusion or exclusion path constraints. From top to bottom: (i) Exclusion constraints are a minimal abstraction for preventing unrealistic assumptions, (ii) inclusion constraints provide a way to represent whether a proper baselines are used, (iii) exclusion constraints ensure that prohibitively expensive execution plans are avoided, and (iv) inclusion constraints are a minimal representation of ensuring implementation plans are detailed, not vague.

| Utility Constraint | Inclusion | Exclusion | Corresponding Failure Mode |
|---|---|---|---|
| Realistic Assump. | ✗ | ✔ | Unrealistic assumptions |
| Ensure Baseline | ✔ | ✗ | Missing or weak baselines |
| Resource Constraints | ✗ | ✔ | Prohibitively expensive execution plans |
| Detailed plan | ✔ | ✗ | Vagueness on implementation details |

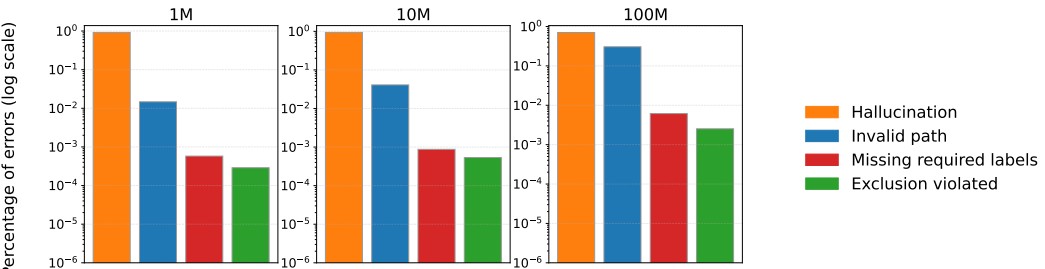

Figure 5: **The distribution of error types on the combinatorial creativity task**. This plot shows the proportion of error types among the creative artifacts that failed to satisfy the utility predicate (term 3 in Definition 5), plotted on a log-scale.

by several orders of magnitude compared to other error types, showing that smaller models mostly fail by producing structurally invalid outputs. However, at the 100M scale, hallucinations decline sharply and "invalid path" errors rise to become nearly equal in frequency. Even though scaling can reduce obvious, superficial errors (e.g., ungrammatical sentences, invalid tokens), deeper problems related to logical inconsistency still remain. As a result, larger models may appear more creative superficially, but their utility errors become subtler and more semantic.

## 6   LIMITATIONS AND CONCLUSION

While our work offers a promising theoretical framework for studying creativity, and our results offer exciting insights into the architectural choices that affect creativity, several limitations remain. Notably, we restricted our focus only to combinatorial creativity (CC), neglecting Boden (2004)'s other two forms (see Appendix C for additional commentary on this). Next, our empirical results relied on synthetic data, which may not be fully representative of the complexity of real-world data encountered in creative domains. Lastly, due to limited compute, we were only able to study up to 100M parameter models, whereas modern foundation models are well into the billions of parameters. Nevertheless, the generality of our framework means it is flexible enough to apply to real-world data, and future studies with access to more compute can explore the scaling behavior beyond the 100M cliff. Together, our conceptual framework and empirical findings offer a new pathway for understanding and improving the creativity of modern AI models, bridging the gap between human and machine intelligence.

## 7   REPRODUCIBILITY STATEMENT

To ensure reproducibility of results, we provide the source code used to obtain the experimental results. In Appendix B, to further support reproducibility of efforts, we provide additional details regarding dataset construction, training, and tokenization.

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

## A RELATED WORK

### A.1 OPEN-ENDED ALGORITHMIC TASKS

LLMs have been increasingly evaluated on open-ended tasks, since open-endedness is seen as a prerequisite for AGI or ASI (Hughes et al., 2024). Khona et al. (2024) use graph pathfinding tasks to study stepwise inference, finding a *diversity-accuracy tradeoff* when varying sampling temperature, as well as a *simplicity bias*, where models choose shortest paths when there are many possible paths. Though their pathfinding task is structurally similar to our combinatorial creativity setting, their task

does not capture creativity since it does not measure degrees of novelty or utility. Focused explicitly on creativity, Nagarajan et al. (2025) recently proposed a suite of open-ended, algorithmic tasks designed to serve as a minimal abstraction of combinatorial and exploratory creativity abilities. Our framework extends theirs by permitting structurally novel artifacts and enabling evaluation of degrees of novelty and utility for individual artifacts.

## A.2 MECHANISTIC UNDERSTANDING OF CREATIVITY IN LLMs

Peeperkorn et al. (2024) have investigated the impact of the temperature parameter on creativity in narrative and story generation. They found a weak positive correlation between temperature and novelty and a negative correlation between temperature and coherence. Interestingly, the authors argued that this suggested a tradeoff between novelty and coherence, which is analogous to the novelty-utility tradeoff observed in this paper. More recently, Morain and Ventura (2025) investigated the impact of prompt engineering techniques on creativity in four prompt domains: joke, poem, six-word story, and flash fiction. They found that "more sophisticated prompting techniques like OPRO and CoT do not produce artifacts of significantly higher quality, novelty, or creativity compared to basic prompting approaches" (p. 9). Lastly, Nagarajan et al. (2025) studied the impact of pre-training objective (next-token prediction versus multi-token prediction) on minimal, algorithmic tasks for combinatorial creativity, finding that multi-token prediction led to increased creativity.

# B ADDITIONAL EXPERIMENTAL DETAILS

## B.1 DATASET CONSTRUCTION

We start with a synthetic graph $\mathcal{G} = (\mathcal{V}, \mathcal{E})$, which serves as the ground-truth "conceptual space". This graph is designed to be large enough to support a rich variety of combinatorial paths, yet sparse enough to make pathfinding a non-trivial challenge. The set of vertices, $\mathcal{V}$, represents the atomic concepts within our synthetic world. We define each node as a unique three-letter capitalized string. This procedure yields a total of $|\mathcal{V}| = 26^3 = 17,576$ distinct nodes, ranging from AAA to ZZZ. The set of edges, $\mathcal{E}$, represents the relationships between these concepts. Crucially, each undirected edge $(u, v) \in \mathcal{E}$ is assigned a label $l$ randomly chosen from the 26 lowercase English letters. These labels are fundamental to our task, as they form the vocabulary for constructing the creative artifacts that our models will be trained to generate.

To create a graph with a controlled level of connectivity, we construct it as an Erdős-Rényi-like random graph. Specifically, we randomly sample node pairs without replacement until we form a graph with an average node degree of approximately six. This results in $|\mathcal{E}| = \text{round}(\frac{1}{2} \times |\mathcal{V}| \times \texttt{avg\_degree}) = \text{round}(\frac{1}{2} \times 17,576 \times 6) = 52,728$ edges. The final graph is stored as a list of edge tokens, where each token is a string concatenation of its source node, label, and destination node (e.g., AAAbCCC).

From the base graph $\mathcal{G}$, we generate a large dataset of query-path pairs for training and evaluation. Each pair consists of a *query*, which specifies a pathfinding problem, and a *path*, which is a valid solution. The queries are designed to vary in difficulty along several axes, allowing us to systematically probe the models' combinatorial abilities.

A single data point is a tuple $(Q, P)$, where $Q$ is the query and $P$ is the ground-truth path. A query $Q$ is defined by a start node $u \in \mathcal{V}$, an end node $v \in \mathcal{V}$, an *inclusion set* $I \subseteq \Sigma_L$, and an *exclusion set* $X \subseteq \Sigma_L$, where $\Sigma_L$ is the set of all 26 lowercase edge labels. A path $P$ is a labeled walk of length $k$, represented as a sequence of nodes and labels $(v_0, l_1, v_1, \ldots, l_k, v_k)$ such that:

1. The path starts at $u$ and ends at $v$: $v_0 = u$ and $v_k = v$.

2. Each step is a valid, labeled edge in the graph: for all $i \in \{1, \ldots, k\}$, $(v_{i-1}, v_i)$ is an edge with label $l_i$.

3. All labels from the inclusion set are used: $I \subseteq \{l_1, \ldots, l_k\}$.

4. No labels from the exclusion set are used: $X \cap \{l_1, \ldots, l_k\} = \emptyset$.

**Training Set Generation** The training set is designed to be large and diverse, providing broad coverage of the graph and various constraint types. Generation proceeds in two stages:

1. **Edge Coverage:** To ensure the model is exposed to every single-step relationship in the graph, we first create a set of simple 1-hop problems. For each edge $(u, l, v) \in \mathcal{E}$, we generate two training instances: one for the path from $u$ to $v$ with inclusion set $I = \{l\}$, and one for the path from $v$ to $u$ with $I = \{l\}$.

2. **Randomized Exploration:** We then generate a large corpus of additional training examples. For each example, we sample a random $(u, v)$ pair and random constraint sets $I$ and $X$. The sizes of these sets are drawn from a geometric distribution to favor simpler queries while still providing a long tail of complex problems. We then execute a constrained BFS to find a valid path up to a maximum length of $h_{\max}^{\text{train}} = 10$.

To ensure a fair evaluation, we enforce a strict holdout policy: any $(u, v)$ node pair that appears in the evaluation set is forbidden from appearing in the training set.

## B.2  TRAINING AND TOKENIZATION

**Hyperparameter Choice**    In line with findings from scaling law research, we adopt a size-dependent learning rate schedule. Models within each parameter bucket (1M, 10M, 100M) are assigned a specific learning rate that decreases with model scale, ensuring that each model is trained under near-optimal conditions and facilitating fair comparisons across sizes. We use the AdamW optimizer with a cosine learning rate decay and a brief warmup period. All models are trained for a fixed 16 epochs to observe the full learning trajectory.

**Pre-Training and Tokenization**    We employ a standard GPT-2 architecture, which learns to predict the next token in a sequence given the preceding ones. The task is framed as conditional generation: the model is given a query $Q$ as a prompt and must generate the corresponding path $P$. To achieve this, we use a custom tokenizer tailored to our conceptual graph. The vocabulary consists of atomic units representing the graph's components: three-letter uppercase tokens for each node, single lowercase letters for edge labels, and special characters for syntax and control (e.g., ':', '[', ']', '<eos>'). This design forces the model to treat concepts as indivisible units, directly aligning with our theoretical view of combinatorial creativity as the recombination of known concepts. All models are trained from scratch on our generated dataset using a standard causal language modeling objective with a cross-entropy loss. The loss is only computed on the path tokens; the query tokens are masked out, conditioning the model without providing supervision for query generation.

**Evaluation**    Model performance is evaluated at the end of each training epoch. We use greedy decoding to generate a single path for every problem in our structured evaluation set.

## C  BROADER IMPACT AND FUTURE WORK

**Evaluating Diversity**    Large-scale empirical studies have discovered that LLMs struggle to produce diverse outputs on scientifically creative tasks (Si et al., 2024). While the algorithmic creativity measure in Nagarajan et al. (2025) ignores degrees of novelty and utility for individual artifacts, it does evaluate the diversity of a large number of outputs, which is one aspect we ignore. Future work can extend the framework introduced in this paper by incorporating diversity as well.

**Scaling Behavior for Exploratory and Transformational Creativity**    Among the three forms of creativity defined by Boden (2004), we only study the combinatorial form. Future work can study the scaling behavior of exploratory and transformational creativity. In particular, it is also worthwhile investigating to what extent LLMs suffer from novelty-utility tradeoffs in exploratory and transformational creativity as well. The transformational creativity frameworks in Thagard (2018) and Schapiro et al. (2025) can serve as a conceptual and mathematical foundation for this line of inquiry.

## C.1  AVENUES FOR IMPROVING MODEL CREATIVITY

**Pre-Training Objective**    Skepticism over the conventional pre-training objective for Transformers, next-token prediction (NTP), has begun to accumulate over the past few years. Bachmann and

Nagarajan (2024) demonstrated the inability for teacher-forcing, NTP training to solve a very simple pathfinding task called *path-star*. In the context of creativity, Nagarajan et al. (2025) later found that multi-token prediction (MTP) led to increased algorithmic creativity on two minimal combinatorial creativity tasks. Recently, token order prediction (TOP) has been proposed to remediate some of the challenges of MTP, finding improved scaling behavior over both NTP and MTP (Zuhri et al., 2025). A promising future direction to explore is the effect of pre-training objective on combinatorial creativity.

**Democratizing Creative AI Through Inference-Time Techniques**   Given the scale-invariant nature of the novelty-utility tradeoff, alternative strategies beyond parameter scaling become crucial for improving creative capabilities, particularly for resource-constrained settings. Recent work by Shashidhar et al. (2023) demonstrates that domain-agnostic self-refinement can yield substantial improvements for smaller models, achieving up to 25.39% improvement on high-creativity, open-ended tasks through iterative self-critique. This is particularly relevant to our findings: if the fundamental creativity constraints persist across scales, then inference-time techniques like self-refinement, which require no additional training, offer a promising path for democratizing access to creative AI capabilities. Rather than requiring massive computational resources to train ever-larger models that still face the same novelty-utility tradeoff, practitioners could leverage smaller, more accessible models enhanced with refinement strategies.

**Architectural Innovations**   The failure modes of LLMs (e.g., frequent errors in responding to simple questions like "How many R's are in strawberry?" or "Is 9.11 or 9.9 bigger?") have prompted many to explore alternative architectures beyond the standard Transformer (Vaswani et al., 2017). Energy-based Transformers (Gladstone et al., 2025) (EBTs) have recently been explored to improve System-2 thinking and generalization as a whole. As Energy-Based Models have demonstrated promising compositional generalization abilities (Du et al., 2023), and compositional generalization overlaps heavily with combinatorial creativity, EBTs could offer promising capabilities for creativity.

