# OpenReview forum: "Combinatorial Creativity: A New Frontier in Generalization Abilities"
_ICLR.cc/2026/Conference — Submitted to ICLR 2026_

### Official Review · Reviewer_vqVz · 2025-10-31

**Soundness:** 2
**Presentation:** 2
**Contribution:** 2
**Rating:** 4
**Confidence:** 4

**Summary:**

This paper seeks to formalize combinatorial creativity as a distinct generalization capability and introduces a framework to evaluate creativity as a continuous trade-off between novelty and utility in structured conceptual spaces. The authors operationalize creativity via graph-based constrained path-generation tasks, paired with explicit novelty and feasibility scoring, enabling systematic measurement beyond binary correctness. Using this setup, they conduct controlled scaling experiments across Transformer architectures with varying parameter sizes and uncover two key findings: (i) creativity does not scale monotonically with size but has optimal depth-width ratios, and (ii) a novelty–utility trade-off exists even for bigger models. The work provides the empirical evidence that creative capability saturates beyond simple scaling.

**Strengths:**

Timely contribution to the recent area of improving creativity in LLMs, with new creativity evaluation metrics.

Findings here can help practitioners choose model hyperparameters for optimal creativity.

**Weaknesses:**

Focuses on a rather limited aspect of creativity (Combinatorial Creativity) which is already included in other work (e.g. https://arxiv.org/pdf/2504.15266). There is also a lack of comparison or description of the difference between these similar works with proposed approach.

Lack of in-depth analysis of observations and findings here, for example, why is there a depth-width trade-off for creativity.

**Questions:**

What is the key difference between the proposed methods and other recent and similar work (e.g. https://arxiv.org/pdf/2504.15266)?

Comment:
Figure 1 & 2: I would suggest the authors to use these figures to more clearly illustrate the overall concepts of the contribution. It is unclear what message the current illustration is trying to convey.

---

### Official Review · Reviewer_ryYk · 2025-10-31

**Soundness:** 2
**Presentation:** 2
**Contribution:** 2
**Rating:** 4
**Confidence:** 4

**Summary:**

This paper introduces a formal framework to evaluate "combinatorial creativity" (CC) in LLMs, distinguishing it from compositional generalization (CG). The authors propose an open-ended, algorithmic task where creativity is modeled as constrained pathfinding in a large, synthetic "conceptual space" (a graph). The framework provides continuous metrics for "novelty" (based on path length and label rarity) and "utility" (based on satisfying inclusion/exclusion constraints). The authors then empirically test decoder-only Transformers (based on a GPT-2 architecture) at 1M, 10M, and 100M parameter scales. Key findings include: (1) an optimal, non-monotonic width-to-depth ratio for creativity at a fixed parameter budget; and (2) a persistent novelty-utility tradeoff that does not diminish with scale, which the authors propose as a potential explanation for the "ideation-execution gap" in scientific idea generation.

**Strengths:**

1. The paper provides a mathematically precise framework (Definitions 1-6) for defining and measuring one specific view of combinatorial creativity.
2. The initial sections (Introduction, Background) provide a good survey of creativity and clearly situate the work against compositional generalization (Table 1).
3. The architectural sweep (Figure 3) is systematic for the defined task, and the error analysis (Figure 5) provides some insight into failure modes on this task.

**Weaknesses:**

1. The paper's primary flaw is its severe scale limitation. Experiments top out at 100M parameters using a GPT-2-like architecture. There is no evidence that these findings can scale up to the performance of modern, multi-billion-parameter large language models, especially those that exhibit 'emergent abilities' not present at the 100M scale. This failure to test at a relevant scale means the framework's applicability to the models the community actually uses is unproven, which severely undermines the paper's contribution.
2. The paper provides zero evidence that performance on this synthetic, semantically-vacant graph task correlates with any real-world creative ability. This "toy" task setup severely limits the generalizability and impact of all findings.
3. The key takeaways, such as the existence of an optimal E/L ratio for a 100M toy model and the persistence of a tradeoff at this small scale, are simply not interesting or impactful conclusions for a top-tier conference. The payoff is not worth the setup.
4. The paper is rife with over-claiming.
  - Claiming this 100M-parameter-limit study "casting doubt on the long-term creative potential of LLMs" is an enormous, unsupported extrapolation.
  - Claiming to explain the "ideation-execution gap" via the simplistic mapping in Table 2 is hollow.
5. The novelty metric (Eq. 2) hard-codes "longer is more novel," which is debatable. The use of an undirected graph to "prevent the reversal curse" feels like an artificial simplification.
6. The paper evaluates creativity using greedy decoding, focusing on a single "best" output. This ignores diversity, a fundamental component of creativity.

**Questions:**

*Also see weakness above

1. The paper's framework is validated only on 100M GPT-2 models. Can you provide any evidence or strong theoretical argument for why this framework and its findings (e.g., optimal E/L ratios, tradeoffs) would be relevant to evaluating modern, multi-billion-parameter foundation models (e.g., Llama 3.1-70B)?
2. Could you comment on the apparent disconnect between the ambitious, broad theoretical framework for 'creativity' and the very narrow, specific empirical study on an outdated architecture? The payoff (e.g., "an optimal E/L ratio exists") seems uninteresting and incommensurate with the grand setup.
3. The novelty metric (Eq. 2) assumes longer paths ($h$) are more novel. How sensitive are your findings to this choice? What happens if you define novelty as an inverse function of path length, rewarding "elegant shortcuts"?
4. Why did you not attempt to validate this framework on even one real-world, non-synthetic task? Even a simple task on a real knowledge graph (e.g., ConceptNet) would make the claims much more convincing.

I want to emphasize that I agree with the overall research direction, which I find valuable. I strongly encourage the authors to address these questions in their rebuttal. Thank you for your work!

---

### Official Review · Reviewer_mxJw · 2025-10-31

**Soundness:** 2
**Presentation:** 3
**Contribution:** 2
**Rating:** 4
**Confidence:** 3

**Summary:**

This paper explores combinatorial creativity (CC) as a new frontier in the model of generalization behavior. The authors discuss various forms of creativity and compare CC with other types of generalization and other CC frameworks from various aspects, e.g., open-endedness, degree of novelty, and degree of utility, where the proposed CC framework measures the creativity based on sampling from the joint distribution of novelty and utility. The authors then experiment with the framework with the GPT-2 decoder and compare the architectural creativity across models from 1M to 100M scales. Results show the relevance between creativity scores and model architecture design, the trade-off between novelty and utility, and the potential ideation-execution gap. With the experiments and analysis, the authors argue that the proposed framework can help understand the creativity of modern AI models.

**Strengths:**

1.	This paper is well-written and easy to follow, with clear motivation on the framework and discussions.
2.	The discussion of creativity (Section 2.1 and Section 2.2) is interesting, covering classic and modern frameworks.
3.	This work provides a systematic framework for quantifying the degree of novelty and utility, which can be used for automatic system-level evaluation.

**Weaknesses:**

1.	The discussion is missing a line of recent literature on applying combination creativity for ideation. Please check: Scideator (https://arxiv.org/abs/2409.14634), CHIMERA (https://arxiv.org/abs/2505.20779), and Ramon Llull’s thinking machine (https://arxiv.org/abs/2508.19200)

2.	The authors claim that the findings provide insights for the ideation-execution gap proposed in Si et al. (2025). However, the original gap was describing the quality between the ideas proposed by the frontier language models and the execution done by real human researchers, with the potential help of code agents. It is hard to draw transferable insights of the relevance between the failure mode at the ideation execution experiments and the current framework with 100M scale training (and token-level definition of  novelty)

3.	Following point 2, the current experiment settings and scales limit the generalization and applicability of findings to the real tasks requiring novelty (e.g., ideation). (1) The random graph task (introduced in Section B.1) is too ad-hoc and hard to generalize to natural language tasks (since the tokens and n-grams are innately relevant). To make my comment concrete, please check A Taxonomy of Transcendence (https://arxiv.org/pdf/2508.17669), where the authors construct a knowledge graph-based simulation to assess the potential generalization behavior, where the fact-based node construction can potentially provide extended insight to general creativity discovery; (2) the scales of the models used also limit the strengths of the claim on the scaling behavior. As a common practice to show the scaling behavior, the authors should train a series of models following the chinchilla scaling laws (https://arxiv.org/pdf/2203.15556) to hedge the confounders, such as over-/under-training, i.e., train and test a series of models following their estimated optimal training flops.

**Questions:**

1. Most of the experimental settings are introduced in Appendix B. Is it possible to adjust the flow to cover at least some details in the main paper?

2. Figure 4 seems to provide limited insight. It seems intuitive that the novelty drops with more utility constraints. Is it possible to consider more conditions for the visualization, e.g., the complexity of the constraints?

---

### Official Review · Reviewer_fTYK · 2025-11-01

**Soundness:** 2
**Presentation:** 3
**Contribution:** 2
**Rating:** 4
**Confidence:** 2

**Summary:**

This paper introduces a formal framework for evaluating combinatorial creativity (CC) in AI systems, modeling it as path-finding in conceptual graphs with logical constraints. Through systematic experiments on decoder-only Transformers (1M-100M parameters), the authors establish: (1) scaling laws for creativity, (2) optimal architectural configurations (depth ~8 layers, E/L ratio 200-300 for fixed budgets), (3) a fundamental novelty-utility tradeoff that persists at scale, and (4) error analyses connecting to the ideation-execution gap in LLM-generated scientific ideas.

**Strengths:**

- First formal framework for combinatorial creativity in AI with explicit, measurable novelty (Definition 4) and utility (Definition 5) functions. Established with interesting connection with problems in graph-theory.

- Systematic design of experiments: data construction reflects the task design, and exhaustive hyper parameter search to find the optimal model structure.

**Weaknesses:**

1. **Lack of Semantic Grounding:** The graph in the training data is synthetically generated with random connectivity. Edge labels are also randomly assigned across many node pairs without semantic constraints. Real creativity involves understanding semantic relationships between two nodes (knowledges, facts, etc.), and real edges connecting two nodes may not have exactly the same label (e.g., can be paraphrased). So it is unclear whether findings transfer to real conceptual spaces where semantic constraints matter.

2. **Edge Label Space Design:** Edge label vocabulary (~26 labels) is much smaller than node space. From a graph-theory perspective, the query complexity can be much lower than expected. It is also unclear whether the optimal width/depth ratios are artifacts of this design choice. The authors should explore when the edge labels are expanded, how will the optimal trade-off change.

3. **Presentation Issues:** For example, inclusion/exclusion constraints $(I, X)$ are introduced in Definition 3 before their purpose is explained (Definition 5), and such explanation is implicit. It is better to re-mention $x=(u, v, I, X)$ in Definition 5.

**Questions:**

Aside from the concerns raised in the Weaknesses part, I have the following questions:

1. In Definition 5, utility scales by number of constraint labels ($|I|$, $|X|$) rather than number of affected edges. Should they be replaced with the actual number of edges? If label 'a' appears in $1000$ edges but label 'b' in $10$ edges, excluding 'a' should be harder than excluding 'b'.

2. In Definition 4, what is $w_{l_i}$? Is this the portion of label $l_i$ in the connecting edges of node $v_{i - 1}$? Also, since for a node $v_{i - 1}$, there can be multiple edges with the same label $l_i$, choosing the next node $v_i$ is also not deterministic. Shall it be considered into $S(P)$ as well?

3. Section 3.3: How are "utility levels" formally defined? Is level $l = |I| + |X| + 1$?

---

### Author Response · Authors · 2025-12-03
**Rebuttal**

# Rebuttal:
We thank the reviewers for their detailed insights. Below, we address the main reviewer concerns in detail:
- **(fTYK-W1, mxJw-W3, ryYk-W2, ryYk-Q4, ryYk-Q1: Semantic grounding/correlation to real-world creativity/limited scale):** Reviewer fTYK questioned “whether findings transfer to real conceptual spaces where semantic constraints matter.” Synthetic tasks are common in generalization/creativity work (PCFG compositionality, synthetic arithmetic, Nagarajan & Wu’s testbed, etc.), where one aims to abstract and compress the essence of a cognitive ability. This is especially important for creativity, which has historically been conveyed in predominantly qualitative terms, rather than operationalized in quantitative mathematical or algorithmic frameworks. Our formalism draws upon numerous sources in cognitive science and psychological literature. The ability to fully isolate structure to probe architectural properties or compute tradeoffs is, in fact, the key strength of minimal, algorithmic tasks like ours, or the earlier framework we build off in  [1], which won an outstanding paper award at ICML 25. Our paper meaningfully extends the CC tasks in [1] by allowing structurally novel artifacts, measuring degrees of novelty, and measuring degrees of utility, which are all integral aspects of CC evaluation overlooked by [1]. Even though our experiments are on an older architecture (GPT-2), the same computational building blocks (Transformer, Attention mechanism) remain in place today. Our findings, like those of [1], should best be thought of as studying the relationship between fundamental architecture choices and creativity.
- **(vqVz-W1, Comb creativity is too limited, needs comparison with Nagarajan & Wu tasks in [1]):** First, combinatorial creativity is not a limited aspect of creativity; in fact, it is perhaps the single most studied form of creativity, with an extremely rich literature across disciplines (philosophy of mathematics, cognitive science, computational creativity, psychology) that dates back to the early 20th century [2]. This makes it well worth studying, and highly applicable to almost any creative act. Second, we have added a detailed comparison within the tasks in Nagarajan & Wu [1] in Section 3.5.
- **(ryYk-W1: Limited scale):** Reviewer ryYk argued “the paper's primary flaw is its severe scale limitation. Experiments top out at 100M parameters using a GPT-2-like architecture… this failure to test at a relevant scale means the framework's applicability to the models the community actually uses is unproven, which severely undermines the paper's contribution.” We agree with ryYK and, in our revisions sections, we significantly modify the scope and claims of the work to quell these overclaims and concerns.
- **(mxJw-Q2, ryYK-3: Utility constraints and the novelty-utility tradeoff):** Per mxJw-Q2, regarding understanding the novelty-utility tradeoff as a function of the complexity of the constraints, we note that Figure 4 (and Definition 6, which Figure 4 is based off of) naturally takes the complexity of the constraints into account.
- **(mxJw-W2, the ideation-execution gap):** Although the original gap was describing the quality between the ideas proposed by the frontier language models and the execution done by real human researchers, we explain in Table 2 how our inclusion/exclusion constraints can serve as minimal abstraction of the failure modes encountered in scientific ideation using LLMs. These are the same failure modes which led to the ideation-execution gap in the first place.
- **(fTYK-W2: Edge space design):** The label space smaller than node space is realistic, since relation types (causes, kind of, type of, etc.) are usually far fewer than the number of entities or concepts in a domain. Even though the query complexity can be much lower than expected, our controllable creativity evaluation in Definition 6 naturally takes this into account. Lastly, we expect the optimal width/depth ratios to be artifacts of this design choice, but our goal was to verify the existence of such a tradeoff, not pinpoint its exact location. Due to limited compute, we feel this is a reasonable contribution.
- **(ryYk-Q3, ryYk-Q5: Novelty metric):** It is standard practice in cognitive neuroscience, psychology, and creative literature to measure novelty using semantic distance [3]. The suggestion that creativity evaluations might want to reward simplicity is indeed correct, but typically this is evaluated as a component of utility [4]. One could already interpret our exclusion constraints as, e.g., enforcing simpler execution plans for scientific idea generation (see Table 2).

## References
- [1] https://arxiv.org/abs/2504.15266
- [2] Boden, Margaret A. The creative mind: Myths and mechanisms. Routledge, 2004.
- [3] https://arxiv.org/abs/2010.07126, https://computationalcreativity.net/iccc20/papers/131-iccc20.pdf
- [4] Thagard, P. (1988). Computational philosophy of science. MIT press.

---

### Author Response · Authors · 2025-12-03
**Revisions**

# Revisions
Taking reviewer feedback into account, we have made the following modifications to the paper:
- **(ryYk-1: Refining scope, removing overclaims):** Most of the reviewers were (rightfully) concerned with the magnitude of claims made in the paper (i.e., “casts doubt on the long-term creative potential of LLMs” or “explains the novelty-utility tradeoff”), given that experiments were only conducted on models up to 100M parameters. In light of this, we have refined the presentation of the paper to better reflect the scope and contributions. Namely, the abstract, introduction, and discussion have all been modified to emphasize that our framework and results can serve as a starting point for understanding and improving the creativity of frontier-size models, but to simultaneously acknowledge the limited scale of our findings.
- **(vqVz-Q1: Detailed comparison with [1]):** Per vqVz’s Q1, we have added a detailed comparison between our task and framework with those of [1] in Section 3.5.
- **(fTYK-Q1, fTYK-Q2, fTYK-Q3: Definitional/presentation issues):** Regarding fTYK’s Q1 about the frequency of edge labels and their role in utility, we note that the frequency of edge labels is directly used to calculate novelty, so this is already incorporated into the existing framework. However, per fTYK’s Q2, we have added additional clarification to Definition 3 so that inclusion and exclusion sets are explained in detail before their formal definition is given in Definition 5. Also per fTYK’s Q2, the non-determinism in edge labels is precisely as intended and already included in the S(P) calculation. We have fixed Definition 4 to explain more clearly what $w_{l}$ represents and how it encodes non-determinism, and we have added a paragraph titled “Evaluation Set Generation” at the end of Section 3.3 that formally explains (i) how the evaluation sets are created and (ii) what the utility levels represent (addressing fTYK’s Q3).
- **(mxJw-W1, Additional related work:)** Per mxJw-W1, we have added commentary on the additional related works applying combinatorial ideation for scientific discovery in Paragraph 3 of Section 5.
- **(vqVz-W2: Why there is a depth-width tradeoff):** Per vqVz-W2, we have added commentary in Paragraph 1 of Section 5 providing a high-level explanation for the observed depth-width tradeoff and suggesting that future work can use our work as a starting point to explore this tradeoff in a more fine-grained, mechanistic manner.

We have uploaded a revised version of the paper reflecting these modifications.

---

### Meta-Review · Area_Chair_4AmE · 2026-01-06

**Summary:**

The submission received unanimous scores of marginal reject. While reviewers appreciated the theoretical formalization, they found a critical disconnect between the broad claims and the limited empirical evaluation, which relied on synthetic random graphs and small architectures.

**Reviewer Concerns:**

The authors clarified technical definitions regarding the utility function and inclusion/exclusion sets, added necessary comparisons to related frameworks, and revised the manuscript to tone down overclaims. However, the fundamental critique concerning the validity and scale of the experiments remains unresolved, which in turn fails to demonstrate how the findings transfer to modern foundation models or inform real-world creative tasks.

**Reviewer Scores:**

All reviewers would likely maintain their scores. While the authors' rebuttal improved the paper's precision and reduced overclaims, it did not provide the empirical evidence necessary to validate the framework's relevance. The consensus remains that the work is methodologically sound but empirically insufficient.

---

### Decision · Program_Chairs · 2026-01-26

Reject